# Impact of COVID-19 on the food security and identifying the compromised food security dimension: A systematic review protocol

**Daniel Teshome Gebeyehu**[1,2]*, **Leah East**[1], **Stuart Wark**[3], **Md Shahidul Islam**[1]

**1** School of Health, Faculty of Medicine and Health, University of New England, Armidale, NSW, Australia,
**2** School of Veterinary Medicine, Wollo University, Dessie, Amhara, Ethiopia, **3** School of Rural Medicine,
Faculty of Medicine and Health, University of New England, Armidale, NSW, Australia

* daniel.teshome@wu.edu.et

## Abstract

### Background

Food security is substantially affected directly by COVID-19 and/or indirectly by the measures adopted for the prevention of COVID-19 transmission. The aim of this systematic review is to summarize the impact of COVID-19 on food security and identify the most compromised food security dimension to ease the food security regulators and actors' intervention prioritisation.

### Methodology

Primary research focused on the impact of COVID-19 on food security will be searched from three online databases (PubMed, Web of Science, and Scopus), manually using a google scholar search engine, and studies' reference list were also manually searched. The prevalence of food insecurity in each study and the most compromised food security dimension including their associated factors will be identified. The food insecurity before and after COVID-19 emergence and the status of food security dimension before and after COVID-19 will be compared and interpreted.

### Discussion

The heterogeneity of the studies and the factors for the variability of outcomes will be discussed. COVID-19 had a negative impact on food security if the food insecurity prevalence before the emergence of COVID-19 is less than during the COVID-19 pandemic. Other confounding factors that can contribute to the high food insecurity prevalence like natural disasters, war, and instability will be considered in addition to COVID-19.

### Registration

This systematic review protocol is registered in PROSPERO under the registration number: CRD42022325475.

**Data Availability Statement:** No datasets were generated or analysed during the current study. All

relevant data from this study will be made available upon study completion.

**Funding:** The author(s) received no specific funding for this work.

**Competing interests:** The authors have declared that no competing interests exist.

**Abbreviations: AHRQ**, Agency for Healthcare Research and Quality; **COVID-19**, Coronavirus Disease of 2019; **GRADE**, Grading of Recommendations, Assessment, Development, and Evaluations; **PRISMA**, Preferred Reporting Items for Systematic Reviews and Meta-Analyses; **PROSPERO**, International Prospective Register of Systematic Reviews; **ROBIS**, Risk of Bias in Systematic Reviews (Healthcare.

## Introduction

Food security exists when all people have access to physically, socially, and economically sufficient, nutritious, and safe food that full fills their dietary demands [1]. COVID-19 is causing a devastating multi-sectoral impact on all local, national, continental, and global levels [2–4]. Since its emergence (31 December 2019), COVID-19 is overwhelming the food security of individuals and households [5–8]. COVID-19 is an infectious disease caused by severe acute respiratory syndrome coronavirus two (SARS-COV-2). COVID-19 is affecting food security directly by affecting the food systems and indirectly by undermining the economic productivity of people, job termination, lockdown, and death of the household leaders [9, 10].

Before the emergence of COVID-19, more than 820 million people were exposed to hunger daily and more than two billion people were facing micronutrient shortages each day which intern exposes them to diseases and short life expectancy [11, 12]. Following the COVID-19 pandemic, food security is compromised due to restrictions in the production, processing, and marketing of food [12]. Especially fresh produce like vegetables, fish and meat recorded a high price due to interruptions in distribution channels [13]. Due to COVID-19, 3.3 billion global workforces are at risk of losing their livelihood [14]. The food insecurity due to COVID-19 affects the global society regardless of their economic status [15, 16], but the severity of the food insecurity is substantial in lower income countries than developed one [12].

### Rationale

According to the FAO [1], food security has four dimensions (availability, access, utilization, and stability). Availability of food is the presence of enough food to feed the general population, within a specified boundary (local, national, continental, or global) [17]. To make food secure, food should be both physically and economically accessible to the whole society regardless of their economic status and distance from the food production area [18]. The available and accessible food should be useful that satisfies the diet and nutrient demand of the individual, households, or the general society [17–19]. The available, accessible, and useful food should be sustainable and should persist for an extended period [12, 17, 18]. Some literature studied the impact of COVID-19 on all the food security dimensions [20] and some studied only one or more [21] dimensions. Unless the whole literature conducted on the impact of COVID-19 on food security is summarised, it is impossible to identify which food security dimensions critically affected. Identifying the most compromised food security dimension is helpful to intervene in food insecurity mitigation. The over-arching objective of this systematic review protocol is to summarise the impact of COVID-19 on food security and identify the most compromised/affected food security dimension. The main questions assessed by this systematic review will be:

1. What is the food insecurity prevalence due to COVID-19/restrictions adopted for COVID-19 prevention?

2. Which food security dimension is more affected by COVID-19 prevention restrictions?

## Methodology

This protocol is prepared based on the PRISMA 2020 guidelines (S1 Checklist). The Cochrane handbook for the systematic review of interventions and the Joanna Briggs Institute (JBI) reviewer's manual were also used as references.

## Eligibility criteria

All original studies conducted on the impact of COVID-19 on food security are eligible, but the focus of this literature had to include one or more of the food security dimensions that covers both the food security situations during and before COVID-19 (Table 1). The studies will be included regardless of their study design and all qualitative, quantitative, and mixed method of studies will be considered. The details of the inclusion and exclusion criteria are indicated in Table 1.

## Information sources

Studies on the impact of COVID-19 on food security will be searched from the three online databases (PubMed, Scopus, and Web of Science). The information from these databases will be collected from the date of COVID-19 emergence (31 December 2019) to 12 June 2022. For literature that will not be accessible through database filtration, manual searching will be applied using the google scholar search engine. During manual searching, the snowball searching technique will be applied to access related publications with our literature of interest. In addition to the database and manual searching methods, references of the eligible studies included from manual and database searches will be explored to avoid missing valuable literature.

## Search strategy

To search literature from databases, the combination of keywords that can represent the impact of COVID-19 on food security will be used. The search term for all databases will be *(Impact) OR (Burden) OR (Effect) AND ("COVID-19") OR ("COVID 19") OR ("SARS-COV-2") OR ("SARS COV 2") OR ("Coronavirus disease 2019") OR ("Coronavirus disease-19") OR ("Coronavirus diseases 19") OR ("Severe acute respiratory syndrome coronavirus 2") OR ("Severe acute respiratory syndrome coronavirus-2") OR ("Wuhan coronavirus") OR ("Novel coronavirus 2019") AND ("Food Security") OR ("Food Insecurity") OR ("Food shortage") OR ("Food Security dimension").* The dates from 31 December 2019 to 12 June 2022, the English language, and research article type will be set as a filtering mechanism in database searching. For manual searching, the title (*the impact of COVID-19 on food security*) will be directly written on the google scholar search engine and the list of articles that are not identified by database screening will be included. The references of the included studies will be investigated for checking the availability of unincluded and valuable literature.

## Selection process

Using the pre-decided searching terms, manual and reference searches, DTG will conduct the primary screening followed by independent checking by the other team members (MSI, SW,

**Table 1. Inclusion and exclusion criteria of studies.**

| Inclusion criteria | Exclusion criteria |
| --- | --- |
| Original research publications | Reviews, case studies, governmental and non-government reports, unpublished studies, grey literature, and opinion letters |
| Written in English language | Written in other languages than English |
| Published from 31 December 2019 to 12 June 2022 | Studies published before or after 31 December 2019 to 12 June 2022 |
| Studies that included one or more food security dimension | Studies that lack the concept of at least one or more food security dimensions |
| Studies that cover both before and after COVID-19 pandemic food security assessments | Studies that report only before or after COVID-19 food security situations |

and LE). These consecutive screening and checking activities are for the purpose of ensuring the quality of the selected manuscripts and avoiding missing important literature. If there will be inconsistencies among the reviewers, all the authors will come together and alleviate the differences with discussion or reviewing the full-text article for certainty, but if there will no consensus on the inconsistencies, the issue will be referred to an external reviewer.

## Data collection process

Literature identified from the three databases will be exported to EndNote X9. The data collection will be done on 01 May 2022 for the first time and repeated fortnightly for three times, which will end on 12 June 2022. The primary data collection from all databases, manual and reference searching options will be done by DTG and both databases and google sites will be independently explored by the three review authors (MSI, LE, and SW) to make sure that the collected studies by DTG are correct and accurate. The literature collected from the three databases and manual searches will be deduplicated using EndNote X9 unique identifier. The duplicate studies that are missed by EndNote de-duplication will be removed manually. After the duplicated studies are removed, the less relevant literature will be removed using the title and abstract. Any discrepancies raised among the review authors will be discussed and solved at any stage of the review process.

## Data items

Since the purpose of this review is to analyse the impact of COVID-19 on food security and identify the gap, all original studies that studied the impact of COVID-19 on one or more of the food security dimensions (availability, accessibility, usability, and stability) will be considered. As the four dimensions of food security are interdependent on each other, any studies that were written on the impact of COVID-19 without considering one or more dimensions will be excluded. The literature will be considered regardless of the statistical analysis (descriptive or inferential) and qualitative studies will be considered. The food security gaps will be analysed on the bases of the four food security dimension (Table 2). The studies will be considered, regardless of their study area (local, national, regional, or global). The literature will not be excluded regarding the type of study participants (individuals, certain communities, households, and the general population). The factors for the food insecurity such as COVID-19 restriction, production reduction, job termination, and household head mortality will be synthesised, and the outcomes will be investigated based on Table 2.

## Study risk of bias assessment

The ROBIS tool, which is the University of Bristol's risk of bias assessment tool for systematic reviews was used. It has three phases (assess relevance (optional), identify concerns with the review process, and judge the risk of bias in the review) [22]. The risk of bias will be assessed using the questions under each domain of phase 2. Each question has "*yes*, *probably yes*, *probably no*, *no*, *and no information*" alternatives/choices. Based on the answers to the questions in phase 2, the risk will be measured into "*low or high or unclear*" in each domain of phase 2, and overall review bias in phase 3. As suggested by Whiting *et al*. [22], the risk of bias for each individual domain of phase 2 and the overall review risk of bias will be presented in graph. The risk of bias assessment will be independently performed by all review authors and any discrepancies among authors will be solved by discussion.

**Table 2. The expected outcomes, considerations, comparisons, and interpretations.**

| Expected outcomes | Considerations | Comparisons | Interpretations |
|---|---|---|---|
| **Food security prevalence** | ✓ Higher household food insecurity experience scale<br>✓ Changes in coping strategy index<br>✓ Change in consumers behaviour (shifting the feeding schedule, food type, amount of eating) | Food insecurity prevalence during & before COVID-19 | COVID-19 has a negative impact on food security if the food insecurity prevalence before COVID-19 is less than post-COVID-19 emergence |
| **Food availability** | ✓ Availability of adequate and quality food for everyone<br>✓ Reduced production<br>✓ Change in food trade flow | The food availability during COVID-19 & pre-COVID-19 | The food availability dimension is affected if the food availability is reduced during COVID-19 than before |
| **Food access** | ✓ Presence of the food near to consumers<br>✓ Restricted access to market<br>✓ Unaffordable price<br>✓ Disturbance of distribution channels | Accessibility of food during & after COVID-19 | The food accessibility is affected if it is reduced during COVID-19 than before |
| **Food usability** | ✓ Coverage of micronutrients<br>✓ Dietary satisfaction of consumers<br>✓ Nutritiousness of food items<br>✓ Safeness of food | Usability of food before & during COVID-19 | The food usability is affected by COVID-19 if it is reduced during COVID-19 than before |
| **Food stability** | ✓ Availability of continues food supply<br>✓ Reliable marketing channel<br>✓ Sustainability of food production | Sustainability of food before & during COVID-19 | The food stability is affected by COVID-19 if the food sustainability during COVID-19 is reduced than before |

## Effect measures

The food insecurity prevalence before and during COVID-19 will be compared in each study and the differences will be summarised in the form of percentage. The food security dimension that is mentioned by all or majority of the reviewed studies as affected by COVID-19 will be taken as the most compromised dimension. The food insecurity report because of low production or shortage of food supply will be considered as the COVID-19's impact on the availability of food and the food insecurity because of high price or distribution problem, will be considered as COVID-19 impact on the accessibility of food. If the available food is of inferior quality, unbalanced, and cannot fulfil one's dietary requirement will be considered COVID-19's impact on usability while the irregular availability of food items that cannot sustain for an extended period due to COVID-19 related restrictions will be considered COVID-19's impact on food stability. The general food insecurity and the effect of COVID-19 on each food security dimension will be analysed in both qualitative (rank) and quantitative (percentage) ways.

## Synthesis methods

This systematic review will be aimed at summarizing the impact of COVID-19 on food security and identifying the most compromised food security dimension. As a result, the review will be focused on answering the following two questions:

1. What is the impact of COVID-19 on food security? This question will be synthesized by gathering information on the prevalence of food insecurity from different literature. Since food insecurity may not only be because of COVID-19, associated factors for the food insecurity will also be analysed. For the studies that did not include the associated food insecurity risk factors, the gap will be critically appraised in the result and discussion of the systematic review. The food insecurity before the emergence of COVID-19 and the food

insecurity prevalence during COVID-19 pandemic will be compared and the impact of the pandemic on food security will be interpreted.

2. What is/are the most affected food security dimension by COVID-19? As described by FAO (2006) the four food security dimensions (availability, access, usability, and stability) should be maintained to be food secure. Food insecurity can occur because of COVID-19's effect on one or more of the food security dimensions. After a careful literature review, the COVID-19's impact on each of these dimensions will be graded as high, moderate, and low.

Based on the answer to these two questions, the food policymakers, and governmental and non-governmental organizations can easily intervene for alleviating the most compromised food security dimensions and the whole food security issue will be resolved.

## Reporting bias assessment

Selective reporting of the result could bring a substantial risk in accessing necessary information in the selected publications. The Agency for Healthcare Research and Quality AHRQ tool for evaluating the risk of reporting bias in systematic reviews will be used. Using the checklists in this tool, all review authors will independently assess the risk of bias due to unreported or missed results. All publication, selective outcome reporting, and selective analysis reporting will be assessed based on the checklists in the tool [23]. Any disagreements among the review authors will be resolved by consensuses. The reporting biases will be judged as "suspected" or "undetected" and the reason for each judgement will be elaborated.

## Certainty assessment

The GRADE tool will be used to assess the certainty of evidence. The factors/question we will consider are: how much the findings are aligned with or topic of interest? how much different findings are consistent with each other? how much the publication bias affects our review? what study limitations do the literatures have? and do the literatures contain the targeted research outcomes? Based on the criteria (a large effect, confounding effect, and under-reporting effect) we had set for GRADE domains, the certainty of evidence will be judged as high, moderate, low, and very low. All the review authors will independently conduct the certainty assessment and any discrepancies will be discussed and solved.

## Discussion

As far as we explored all the systematic reviews done before, a systematic review on the impact of COVID-19 on the food security and identifying the compromised food security dimension or its similar is not done yet. Among the multidimensional crises due to COVID-19, food security is one of the most compromised sectors. Summarising different research findings on the impact of COVID-19 on the food security and identifying the most affected food security dimension is the foundation for prioritising interventions and to made evidence-based decision.

The COVID-19 pandemic or measures adopted to prevent its transmission, impacted the food security of humans. Movement restriction, job termination, market closure, disruption of the food market and distribution channels, death of household leaders, reduced food production, and other socio-economic crises are the factors for the high prevalence of food insecurity. As a result of these factors, one or more of the food security dimensions might be affected. and identifying the compromised dimension is the purpose of this review in addition to summarizing the general food security crises due to COVID-19.

To be food secure, safe, nutritious, sustainable, and sufficient food that can be accessible to the whole community without physical and economic barriers, and able to satisfy the cultural and dietary needs of peoples should be fulfilled. That means the four food security dimensions (availability, access, usability, and stability) should be there for the presence of complete food security. The focus of different literature might be on one or more of the food security dimensions. The food security dimension/s that is/are repeatedly reported by different researchers will be identified and will be recommended to different food security actors and regulators for immediate intervention. Not only the food security dimension, but also the contributing factors for the food insecurity will be identified. Confounding factors like war, instability, natural disaster, and pest damages will be considered and elaborated depending on the study area and time of the reviewed studies. As a result, intervention will be proposed on both the compromised food security dimension and the contributing factors.

This systematic review has limitations in including all research articles published in different languages than English. The multisectoral impacts of COVID-19 other than its food security burden are not included in this review. Not only this, but the present review will also not include studies that did not assess both the pre and post COVID-19 emergence food security situations. Literature other than research articles will be excluded. These limitations are recommended to be covered by other reviews of the same author or different review authors. The publication of this protocol can aid other review authors who are interested to review the impact of any disease on food security and its dimensions.

## Publication status

This protocol is prepared for the ongoing systematic review entitled "the impact of COVID-19 on food security and identifying the compromised food security dimension: A systematic review". Any change to this protocol will be updated on the PROSPERO registration database under the registration number: CRD42022325475 and the updated part of the protocol will be published together with the full systematic review.

## Supporting information

**S1 Checklist. PRISMA-P checklists.**
(PDF)

## Author Contributions

**Conceptualization:** Daniel Teshome Gebeyehu.

**Formal analysis:** Daniel Teshome Gebeyehu.

**Investigation:** Daniel Teshome Gebeyehu.

**Methodology:** Daniel Teshome Gebeyehu.

**Resources:** Daniel Teshome Gebeyehu.

**Software:** Daniel Teshome Gebeyehu.

**Supervision:** Leah East, Stuart Wark, Md Shahidul Islam.

**Validation:** Leah East, Stuart Wark, Md Shahidul Islam.

**Visualization:** Leah East, Stuart Wark, Md Shahidul Islam.

**Writing – original draft:** Daniel Teshome Gebeyehu.

   

**Writing – review & editing:** Daniel Teshome Gebeyehu, Leah East, Stuart Wark, Md Shahidul Islam.

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
