## [Decision Letter · Decision Letter 0]

4 Jul 2022

PONE-D-22-13987The impact of COVID-19 on the food security and identifying the compromised food security pillar: A systematic review protocolPLOS ONE

Dear Dr. Gebeyehu,

Thank you for submitting your manuscript to PLOS ONE. After careful consideration, we feel that it has merit but does not fully meet PLOS ONE’s publication criteria as it currently stands. Therefore, we invite you to submit a revised version of the manuscript that addresses the points raised during the review process.

We look forward to receiving your revised manuscript.

Kind regards,

Dragan Pamucar

Academic Editor

PLOS ONE

Journal Requirements:

Reviewers' comments:

Reviewer's Responses to Questions

**Comments to the Author**

1. Does the manuscript provide a valid rationale for the proposed study, with clearly identified and justified research questions?

Reviewer #1: Yes

Reviewer #2: Yes

2. Is the protocol technically sound and planned in a manner that will lead to a meaningful outcome and allow testing the stated hypotheses?

Reviewer #1: Yes

Reviewer #2: Yes

3. Is the methodology feasible and described in sufficient detail to allow the work to be replicable?

Reviewer #1: Yes

Reviewer #2: Yes

4. Have the authors described where all data underlying the findings will be made available when the study is complete?

Reviewer #1: Yes

Reviewer #2: Yes

5. Is the manuscript presented in an intelligible fashion and written in standard English?

Reviewer #1: Yes

Reviewer #2: Yes

6. Review Comments to the Author

You may also provide optional suggestions and comments to authors that they might find helpful in planning their study.

Reviewer #1: 1. L. 26 and other parts of the manuscript; do you want to say google search engine? or google scholar search engine?

2. L. 26-27: replace the “references of included studies from the databases and manual searches” with studies’ reference list were also manually searched.

3. Please shorten the sentences from line 129-133 the shortest and most informative sentence “The four pillars of food security are interdependent on each other. If one or more of these pillars are affected by COVID-19, the whole food security system will be compromised. Identifying the more affected food security pillars is crucial to intervene in food security alleviation. As a result, all studies that were written on the impact of COVID-19 without considering one or more pillars will be excluded”.

4. L. 148: Who is the owner of the ROBIS tool? Better to include it.

5. L. 195: Write the AHRQ in full in first use.

Reviewer #2: Dear Author(s),

I appreciate your effort on preparing this protocol. I recommend the following comments/changes to improve your protocol.

1. Page 2 line 20: “severely” to substantially

2. Page 2 line 24: “Original research publications” to Primary research focused

3. Page 4 line 78-79: “Concept” to ‘focus’, and please paraphrase this sentence in a clear way.

4. Page 5 line 85: delete “main”

5. Page 5 line 94: delete “the proposed”

6. Page 5 line 101: better to delete “original”

7. Page 5 line 105: better to use the word valuable than “eligible”. To say… unincluded and valuable literature.

8. Page 5 line 109: Please replace the phrase “the senior experts” with ‘other team members’

9. Page 6 line 122: “Redundant” to ‘duplicate’

10. Page 6 line 124: “sort out” to ‘removed’

11. Page 6 line 134: What about qualitative studies?

12. Page 6 line 135-137: Better to remove the sentence

13. Page 6 line 138-140: The sentence “To be sure of the content of the literature whether they contain the food security pillars or not, the abstract of the literature will be red before they are excluded as unfit” is repeated.

I wish all the best for your protocol and systematic review publication.

7. PLOS authors have the option to publish the peer review history of their article (what does this mean?). If published, this will include your full peer review and any attached files.

Reviewer #1: No

Reviewer #2: **Yes: **Shaharior Rahman Razu

---

## [Author Response · Author response to Decision Letter 0]

11 Jul 2022

Rebuttal letter is attached as "a response to reviewers".

---

## [Decision Letter · Decision Letter 1]

28 Jul 2022

Impact of COVID-19 on the food security and identifying the compromised food security dimension: A systematic review protocol

PONE-D-22-13987R1

Dear Dr. Gebeyehu,

We’re pleased to inform you that your manuscript has been judged scientifically suitable for publication and will be formally accepted for publication once it meets all outstanding technical requirements.

Kind regards,

Dragan Pamucar

Academic Editor

PLOS ONE

Additional Editor Comments (optional):

Reviewers' comments:

Reviewer's Responses to Questions

**Comments to the Author**

1. Does the manuscript provide a valid rationale for the proposed study, with clearly identified and justified research questions?

Reviewer #1: Yes

Reviewer #2: Yes

2. Is the protocol technically sound and planned in a manner that will lead to a meaningful outcome and allow testing the stated hypotheses?

Reviewer #1: Yes

Reviewer #2: Yes

3. Is the methodology feasible and described in sufficient detail to allow the work to be replicable?

Reviewer #1: Yes

Reviewer #2: Yes

4. Have the authors described where all data underlying the findings will be made available when the study is complete?

Reviewer #1: Yes

Reviewer #2: Yes

5. Is the manuscript presented in an intelligible fashion and written in standard English?

Reviewer #1: Yes

Reviewer #2: Yes

6. Review Comments to the Author

You may also provide optional suggestions and comments to authors that they might find helpful in planning their study.

Reviewer #1: I would like to say thank you for correcting all of my comments. The manuscript is clear, informative, and has scientific contributions to readers now. In this regard, I recommend the manuscript be accepted for publication.

Reviewer #2: Dear author, I appreciate the revisions you have done. Thank you your for resubmission and I wish you all the best.

7. PLOS authors have the option to publish the peer review history of their article (what does this mean?). If published, this will include your full peer review and any attached files.

Reviewer #1: No

Reviewer #2: No

---

## [Editor Report · Acceptance letter]

1 Aug 2022

PONE-D-22-13987R1 

Impact of COVID-19 on the food security and identifying the compromised food security dimension: A systematic review protocol 

Dear Dr. Gebeyehu:

I'm pleased to inform you that your manuscript has been deemed suitable for publication in PLOS ONE. Congratulations! Your manuscript is now with our production department. 

Kind regards, 

on behalf of

Dr. Dragan Pamucar 

Academic Editor

PLOS ONE